# Interventions for treating obstetric fistula: An evidence gap map

**Eugenie Evelynne Johnson** [1]*, **Nicole O'Connor** [1], **Paul Hilton** [2], **Fiona Pearson** [1,3], **Judith Goh** [4], **Luke Vale** [1]

**1** Population Health Sciences Institute, Newcastle University, Newcastle upon Tyne, United Kingdom, **2** Cochrane Incontinence, Population Health Sciences Institute, Newcastle University, Newcastle upon Tyne, United Kingdom, **3** NIHR Innovation Observatory, The Catalyst, Newcastle upon Tyne, United Kingdom, **4** Griffith University School of Medicine, Queensland, Australia

* eugenie.johnson@newcastle.ac.uk

**Data Availability Statement:** All information underlying the findings reported are included in the submitted article and its supplementary information files.

## Abstract

Obstetric fistula is prevalent in low- and middle-income countries, with between 50,000 and 100,000 new cases each year. The World Health Organization aims to eradicate it by 2030 but a clear idea of the extant evidence is unavailable. This evidence map compiled evidence on treatments for obstetric fistula to identify potential knowledge gaps. The protocol for this work was published on the Open Science Framework (DOI: 10.17605/OSF. IO/H7J35). A survey was developed, piloted and distributed online through organisations with an interest in obstetric fistula and snowballing. Results informed the evidence map framework. Searches were run on MEDLINE, Embase, CENTRAL, Global Index Medicus and ScanMedicine on 16 February 2022 to identify potentially eligible systematic reviews, randomised controlled trials, cohort studies and case-control studies. Forward and backward citation chaining was undertaken on relevant systematic reviews and included studies. Studies were screened, coded and assessed for risk of bias by a single reviewer, with a second checking a proportion. The evidence map results were compared to survey results. Thirty-nine people responded to the survey, half of which were clinicians. Of 9796 records identified, 37 reports of 28 studies were included in the evidence map. Many included studies were at some risk of bias; for observational studies, this was predominantly due to lack of controlling for confounders. Most studies (71%) assessed surgical interventions alone. Reporting on other intervention types was limited. Regarding outcome measures most important to survey respondents, 24 studies reported on cure/improvement in obstetric fistula and 20 on cure/improvement in urinary incontinence. Reporting on quality of life, faecal incontinence and sexual function was limited. There is currently little robust evidence to guide patients and practitioners on the most effective treatment option for obstetric fistula. Further research is required to address evidence gaps identified.

**Funding:** This work was supported by Newcastle University as part of Training Fellowships for EEJ and NO'C, Newcastle University as part of a tenured post for LV, and various National Institute for Health Research (NIHR) funding, including the NIHR Innovation Observatory, for FP. Neither Newcastle University nor the NIHR had any role in study design, data collection and analysis, decision to publish, or preparation of the manuscript. PH and JG did not receive funding for this work.

**Abbreviations:** AMSTAR 2, Assessment of Multiple Systematic Reviews 2; CBT, cognitive behavioural therapy; CENTRAL, Cochrane Central Register of Controlled Trials; COMET, Core Outcome Measures in Effectiveness Trials; CONSORT, Consolidated Standards of Reporting Trials; JSON, JavaScript Object Notation; LMICs, low- and middle-income countries; MEDLINE, Medical Literature Analysis and Retrieval System Online; MeSH, Medical Subject Headings; PICO, Population, intervention, comparison, outcome; PRESS, Peer Review of Electronic Search Strategies; PRISMA, Preferred Reporting Items for Systematic Reviews and Meta-Analyses; RCT, randomised controlled trial; STROBE, Strengthening the Reporting of Observational Studies in Epidemiology; WHO, World Health Organization.

## Introduction

Obstetric fistula can be defined as an abnormal opening between a woman's genital tract (vagina, cervix, uterus) and either or both urinary (bladder, urethra, ureter) and intestinal tracts (rectum, anus, perineum). It occurs due to prolonged obstructed labour that results in ischemia and necrosis in pelvic soft tissues [1]. Obstetric fistula is mainly categorised as either: vesicovaginal (between the anterior vaginal wall and posterior bladder); urethrovaginal (between the urethra and vagina); and rectovaginal (between the rectum or colon and the vagina); vesico-uterine (between the bladder and uterus); or ureterovaginal (between the ureter and the vagina) [1].

Obstetric fistula is a major burden for women in low- and middle-income countries (LMICs), particularly in sub-Saharan Africa and South Asia. The World Health Organization (WHO) has estimated that between 50,000 and 100,000 women are affected by new occurrences of obstetric fistula each year [2]. A meta-analysis of national household survey data suggested that the lifetime prevalence of obstetric fistula in Sub-Saharan Africa was three cases per 1000 women of reproductive age, with some countries (such as Uganda, Comoros and Kenya) reporting far higher lifetime prevalence [3]. A systematic review of cross-sectional and cohort studies suggested that prevalence in South Asian regions could be up to 1.2 per 1000 women [4].

The condition has serious physical, mental and social impacts. Women with obstetric fistula invariably experience urinary or faecal incontinence or both; they may additionally suffer from sexual dysfunction, and urinary dermatitis [5, 6]. Subsequently, they may face discrimination in their community and a lack of social support, leading to divorce, ostracism and loss of income or employment [5, 7–9]. Consequently, women can experience mental health difficulties, such as anxiety and suicidal ideation [5–7]. A recent systematic review of cross-sectional and cohort studies noted that the prevalence of depression among women with obstetric fistula over 70% in Ethiopia and Kenya [10].

Interventions for treating or managing obstetric fistula include lifestyle management, catheter insertion, physical therapy and psychological therapies [1, 5]. However, surgical intervention is the primary mode of repairing obstetric fistula. There are several types of surgical procedures for treating obstetric fistula, including native tissue repair, autologous grafts and flaps, urinary diversion and debridement of fistula [1].

WHO's Sustainable Development Goal 3.7 states: "By 2030, ensure universal access to sexual and reproductive health-care services, including for family planning, information and education, and the integration of reproductive health into national strategies and programmes" [11]. Part of this goal includes ending fistula by 2030, with a recent paper suggesting that robust research, particularly surrounding surgical procedures, is needed to achieve this aim [12]. Women with obstetric fistula face other barriers to accessing surgical repair procedures. It has been noted that, although surgery is the primary treatment modality, there is a global shortage of skilled fistula surgeons, meaning repair rates are low [13]. Indeed, a qualitative evidence synthesis identified 86 studies that collectively highlighted lack of facilities and expertise as barriers to obstetric fistula repair [14]. The costs of surgery to women and their families is a further barrier. This can include the costs of treatment itself but there are also costs in travelling to specialist fistula repair centres as well as time away from usual activities [15]. Despite these barriers to surgery, there has been little discussion surrounding non-invasive interventions that could potentially assist women with obstetric fistula in managing their symptoms to date. Consequently, stakeholders need a comprehensive, accessible summary of the literature and its overall quality to help guide future research and policy.

Evidence maps are an innovative visual approach to evidence synthesis that establishes the breadth and depth of evidence on a topic [16, 17]. As the evidence surrounding interventions for obstetric fistula is likely to be broad and disparate [18], conducting the first evidence map of interventions for treating obstetric fistula will visualise and contextualise the current breadth and quality of the evidence on this highly prevalent and often devastating condition. Embedding the perspectives of healthcare practitioners, patients and the public into the evidence through a survey will help identify evidence gaps, which can inform future primary research in priority areas, or commission high-quality systematic reviews.

## Objectives

- To survey stakeholders to help construct a framework for an evidence map;

- To produce an evidence map of individual quantitative primary studies and systematic reviews assessing interventions for treating obstetric fistula; and

- To identify potential areas for further research.

# Methods

## Protocol registration and reporting

The protocol for this review was prospectively registered on the Open Science Framework on 15 February 2022 (DOI: 10.17605/OSF.IO/H7J35) [19]. Differences between protocol and review are detailed in S1 Text. Reporting of this work is in accordance with the Campbell Collaboration reporting standards for evidence gap maps, available in S1 Table [20].

## Survey methods

**Survey development.**   The survey was developed, in Qualtrics, to guide the framework of the evidence map and reporting [21]. The survey was piloted on nine participants (a combination of students and researchers at Newcastle University) alongside a short feedback form. Results of the feedback are shown in Table 1. The survey was adapted based on free-text responses to an open question asking how the survey could be improved. A text version of the final survey can be found in S2 Text.

**Survey distribution.**   The final survey was emailed to several organisations with an interest in fistula, such as charities, on 16 February 2022, asking if they could cascade to mailing lists or anyone who may be interested in responding. Additionally, the survey was distributed via Newcastle University's Health Economics and Evidence Synthesis Twitter account and an advert placed on Cochrane's TaskExchange platform. Clinical advisors (PH and JG) were also

**Table 1. Feedback from pilot survey (n = 9).**

| Question | Completed on computer/laptop (n = 8) | | Completed on mobile phone (n = 1) | |
|---|---|---|---|---|
| | Yes | No | Yes | No |
| Was the introduction to the survey and its purpose understandable? | 8 | 0 | 1 | 0 |
| Do you think the questions were worded in an understandable way? | 8 | 0 | 1 | 0 |
| Do you think the survey was structured well and easy to follow? | 8 | 0 | 1 | 0 |
| Do you think the survey was easy to fill out? | 8 | 0 | 1 | 0 |

asked to snowball the survey to contacts with an interest in the field. The survey remained open until 14 March 2022.

**Survey analysis.**   Survey results were collated within Qualtrics [21], then exported to Excel and tabulated. All responses were analysed; only participants under 18 years of age were excluded. Demographic information such as self-identified role, continent and country of residence were tabulated and are reported narratively.

## Evidence map methods

**Search strategies.**   As suggested by the Campbell Collaboration, development of the search strategy was iterative and went through several cycles of piloting and refinement [17]. Search strategies using MeSH terms and text words were developed for MEDLINE, Embase, CENTRAL and Global Index Medicus. Study design filters for MEDLINE and Embase were sourced from the Ovid Tools & Resources Portal [22]. A search strategy for ScanMedicine, which searches 11 global trial registries, was devised to identify ongoing studies that may be relevant to any potential future iterations of the evidence map but currently do not provide data to inform policy or practice. Search strategies were not limited by date, publication type or language.

Search strategies were peer-reviewed twice by an experienced Information Specialist using the Peer Review of Electronic Search Strategies (PRESS) guideline [23]. Search strategies were adapted and updated according to the recommendations in the assessment. Full search strategies for each database as run are documented in S2 Table.

**Eligibility criteria.**   The eligibility criteria were organised according to the Population, Intervention, Comparison and Outcome (PICO) framework. The evidence map's full inclusion and exclusion criteria are documented in S3 Table.

In brief, RCTs, case-control, prospective and retrospective comparative cohort studies were included. Systematic reviews were included if they contained RCTs and/or other eligible non-randomised studies meeting the eligibility criteria. Eligible participants in studies were women of any age, living in any setting, with fistula described as being of obstetric aetiology (vesicovaginal, vesico-uterine, urethrovaginal, rectovaginal, ureterovaginal). Only studies and systematic reviews where at least 80% of the population had fistula of obstetric aetiology were included to ensure maximum relevance to the population of interest. Articles that described fistula simply as "obstetric", irrespective of more detailed reporting on fistula type or aetiology (e.g. vaginal birth or caesarean) were also included, to ensure breadth. For these studies it was assumed that most participants would have fistula of obstetric origin but were labelled as "not described" when coding for type of fistula (see 'Study selection and data coding').

Eligible interventions included: lifestyle interventions such as skin protection and dietary modification; bladder or ureteral catheterisation; physical therapy such as therapeutic exercise and bladder training; psychological therapies such as cognitive behavioural therapy (CBT); and surgical interventions including native tissue repair, graft repair and tissue flaps, and urinary or faecal diversion procedures. Studies were not excluded based on outcome measures, in accordance with recommendations from the Campbell Collaboration [17]. Books, book chapters, editorial commentaries and letters were excluded.

**Study selection and data coding.**   Titles and abstracts identified by the searches were exported into EndNote for deduplication before being exported to Rayyan for screening [24]. Initially, two reviewers (EEJ and NO'C) independently screened titles and abstracts of a random sample of 10% of records. A single reviewer (EEJ) undertook the remaining 90% of title and abstract screening. Full-texts of potentially relevant titles and abstracts were retrieved and a random sample of 20% screened by two reviewers (EEJ and NO'C) in Rayyan [24]. A single

reviewer (EEJ) screened the remaining 80% of full-texts for eligibility. Disagreements were resolved by discussion or, where necessary, referred to a third party for arbitration (PH). It had been planned to assess additional records in duplicate if more than 10% of decisions were in conflict at any screening stage; as conflicts were below 10% at both title and abstract and full-text stage, single-screening was initiated. Forward and backward citation chaining was conducted by a single reviewer (EEJ) for all included primary studies and systematic reviews to help minimise the possibility of missing any eligible records.

Records not reported in English were not formally included but considered 'Awaiting Classification'; brief details of these were reported in the Results. Similarly, ongoing studies were not reported in the evidence map but details are summarised.

Full-texts of all included studies were exported from Rayyan into EPPI-Reviewer [25]. Within EPPI-Reviewer, a coding tool was developed to extract data from each of the included papers. The tool included all aspects suggested by the Campbell Collaboration [25], Other domains were added for transparency surrounding additional linked publications to the primary record, population, length of follow-up, and space to include full risk of bias assessments. Full details of the coding tool used are documented in S4 Table. Data coding was undertaken by a single reviewer (EEJ) using EPPI-Reviewer [25], with 10% of these records being checked by a second reviewer for accuracy (NOC). Disagreements were resolved by discussion.

**Risk of bias assessment.** Systematic reviews were assessed for risk of bias using the Assessment of multiple systematic reviews 2 (AMSTAR 2) tool [26]. Of the identified primary studies, RCTs were assessed using Cochrane's 'Risk of Bias' tool, case-control studies with and cohort studies with the Joanna Briggs Institute (JBI) checklist for cohort studies [27, 28]. Initially, 20% of all assessments were undertaken by a single reviewer (EEJ) and checked by a second (NO'C), with disagreements resolved by discussion. The remaining 80% of included studies and reviews were assessed by a single reviewer (EEJ). As with study selection, it had been planned to assess more studies for risk of bias in duplicate if more than 10% of decisions were in conflict; as conflicts were below 10%, risk of bias assessment by a single reviewer was initiated.

**Data synthesis.** The unit of analysis for this work was the study-level, as suggested by the Campbell Collaboration [29]. However, it is not possible to merge papers into a single study using EPPI-Reviewer [25]. To avoid "inflation" of data, only the primary paper for each study was imported into EPPI-Reviewer (i.e. the report with the most detail contributing to data coding and risk of bias assessment). However, additional reports of studies or trial registrations were referenced within the map using a specific field within the coding tool.

The completed coding tool was exported from EPPI-Reviewer in a JSON format and imported into the EPPI-Mapper wizard to generate the evidence map. Study design (RCT, non-RCT, systematic review) was the segmenting variable. The evidence map is presented in the Results Section below alongside with a narrative report of key findings, in accordance with Campbell Collaboration guidance [17]. This narrative report accounted for the views of survey respondents, who helped identify knowledge gaps and areas for potential further research important to them. To do this, the results of the evidence map were imported into Excel for each outcome and intervention, then tabulated alongside the survey results to facilitate comparison.

**Sensitivity analysis.** Sensitivity analyses were undertaken to explore the effect of removing studies where there was uncertainty relating to the population type due to poor reporting of fistula type and aetiology, these were coded as 'not described' in the evidence map. To determine how applicable the identified evidence was to women with fistulae of obstetric origin and if the data could be used to assess the effectiveness of different interventions. This was achieved

by filtering the evidence map to include only studies where the proportion of women with obstetric fistulae was 80% or over and reported multiple interventions separately.

## Ethical approval

Ethics for the survey were approved by the Faculty of Medical Sciences Research Ethics Committee, part of Newcastle University's Research Ethics Committee (approval reference: 2211/14370). Respondents to the survey were alerted that by completing and submitting the survey, they were giving informed consent for their data to be used by the research team.

## Results

### Survey results

**Characteristics of survey respondents.** In total, 40 participants began completing the survey. One respondent was under 18 and excluded from the analysis. Of the remaining 39 respondents, three (8%) opened the survey but did not provide further information. Characteristics of survey respondents are presented in Table 2. Most respondents were clinicians (60%), and many either resided in Europe (41%) or Africa (36%). Only four respondents (10%) were patients or members of the public. Of those who identified their role as 'Other' (n = 3; 8%), one was in an administrative position, and two were retired clinicians.

**Results of the survey.** The results of the question asking respondents to confirm which intervention category they believed most important to research into obstetric fistula are summarised in Fig 1. Twenty-three respondents (58.97%) stated that surgical interventions were the most important treatment modality. Accordingly, the rows of the evidence map were structured from top to bottom as: surgical intervention; psychological therapy; catheter insertion; lifestyle management; and physical therapy. However, it is of note that, of the patient respondents (n = 4), physical, psychological, and lifestyle interventions were the only interventions felt to be of importance.

The results of the question asking respondents to confirm which six outcomes they believed most important to research into obstetric fistula are summarised in Fig 2. Improvement in quality of life was the outcome measure most frequently listed as being the most important

**Table 2. Characteristics of survey respondents (n = 39).**

|  | Number | Percentage |
|---|---|---|
| **Role** | | |
| Clinician | 23 | 59% |
| Patient or member of the public | 4 | 10% |
| Researcher | 4 | 10% |
| Clinical Academic | 3 | 8% |
| Other | 2 | 5% |
| Missing | 3 | 8% |
| **Continent of residence** | | |
| Europe | 16 | 41% |
| Africa | 14 | 36% |
| North America | 3 | 8% |
| Latin America | 1 | 3% |
| Asia | 1 | 3% |
| Oceania | 1 | 3% |
| Missing | 3 | 8% |

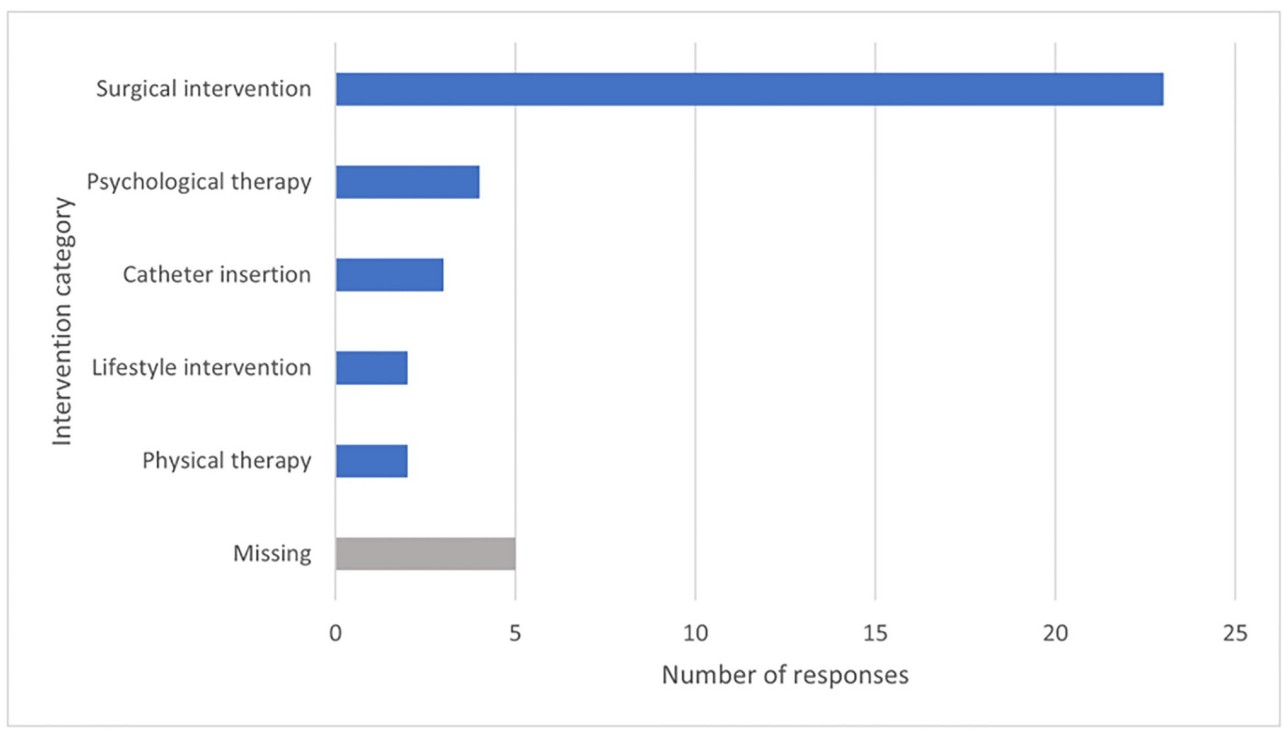

**Fig 1. Summary of survey responses relating to intervention category (n = 39).**

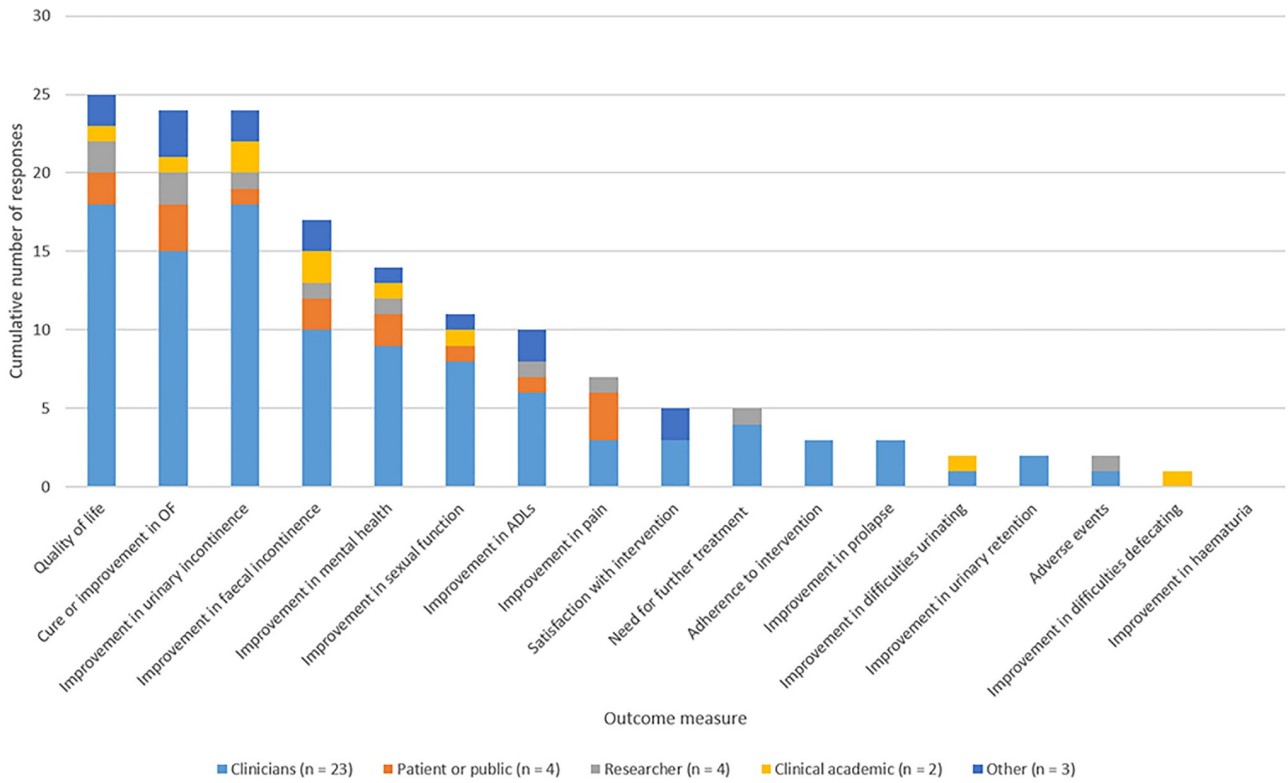

**Fig 2. Summary of survey responses relating to outcomes of interest, separated by respondent role.**

**Table 3. Structure of outcomes for evidence map columns based on survey result.**

| Parent code | Child codes |
|---|---|
| Most important to survey respondents | • Improvement in quality of life<br>• Cure or improvement of obstetric fistula<br>• Improvement in urinary incontinence<br>• Improvement in faecal incontinence<br>• Improvement in mental heath<br>• Improvement in sexual function |
| Moderately important to survey respondents | • Improvement in activities of daily living<br>• Improvement in pain<br>• Patient satisfaction with intervention<br>• Woman's need for further treatment<br>• Adherence to the intervention<br>• Improvement in associated pelvic organ prolapse symptoms |
| Least important to survey respondents | • Improvement in difficulties urinating<br>• Improvement in urinary retention<br>• Adverse events of interventions<br>• Improvement in difficulties defecating<br>• Improvement in haematuria |

outcome measure, identified by 25 respondents. No respondents identified improving haematuria as the most important outcome. Of note, for this question there was less deviation between what different stakeholders believed to be important to research.

In response, the columns of the evidence map were structured according to the scheme outlined in Table 3.

## Evidence map results

**Results of the search.** In total, 9624 records were identified from database searches. After deduplication, 7906 titles and abstracts were screened for eligibility, of which 7343 were excluded. Overall, 563 full texts were sought for retrieval; 39 could not be found. Of 524 full texts retrieved, 428 were excluded. Reasons for exclusion are documented in S5 Table.

From citation chaining and other sources, 172 records were identified, but 120 had either been previously retrieved by the literature searches or were duplicates. One of the remaining 52 records could not be retrieved at full text. Of the 51 remaining records assessed at full-text, 48 were excluded. Reasons for exclusion documented in S5 Table.

In total, 37 reports of 28 studies were included, 57 studies were Awaiting Classification and four studies were ongoing. Six studies were found to have multiple reports; these are detailed in S6 Table. Reasons for placing records in Awaiting Classification and characteristics of ongoing studies are described in S7 and S8 Tables. The flow of literature is shown in the PRISMA diagram in Fig 3 [30].

**Characteristics of included studies.** Characteristics of included studies are summarised in Table 4.

The included primary studies were conducted in: Ethiopia [31–35]; Nigeria [36–40]; Iraq [41, 42]; Pakistan [43, 44]; Benin [45]; Rwanda [46]; the USA [47]; Democratic Republic of Congo [48]; Malawi [49]; Burundi [50]; and Tanzania [51]. Five primary studies were performed across multiple countries [52–56].

**Risk of bias.** Seven RCTs were assessed with the Cochrane 'Risk of Bias' tool, 19 cohort studies were assessed with the JBI Checklist for Cohort Studies, one study was assessed with

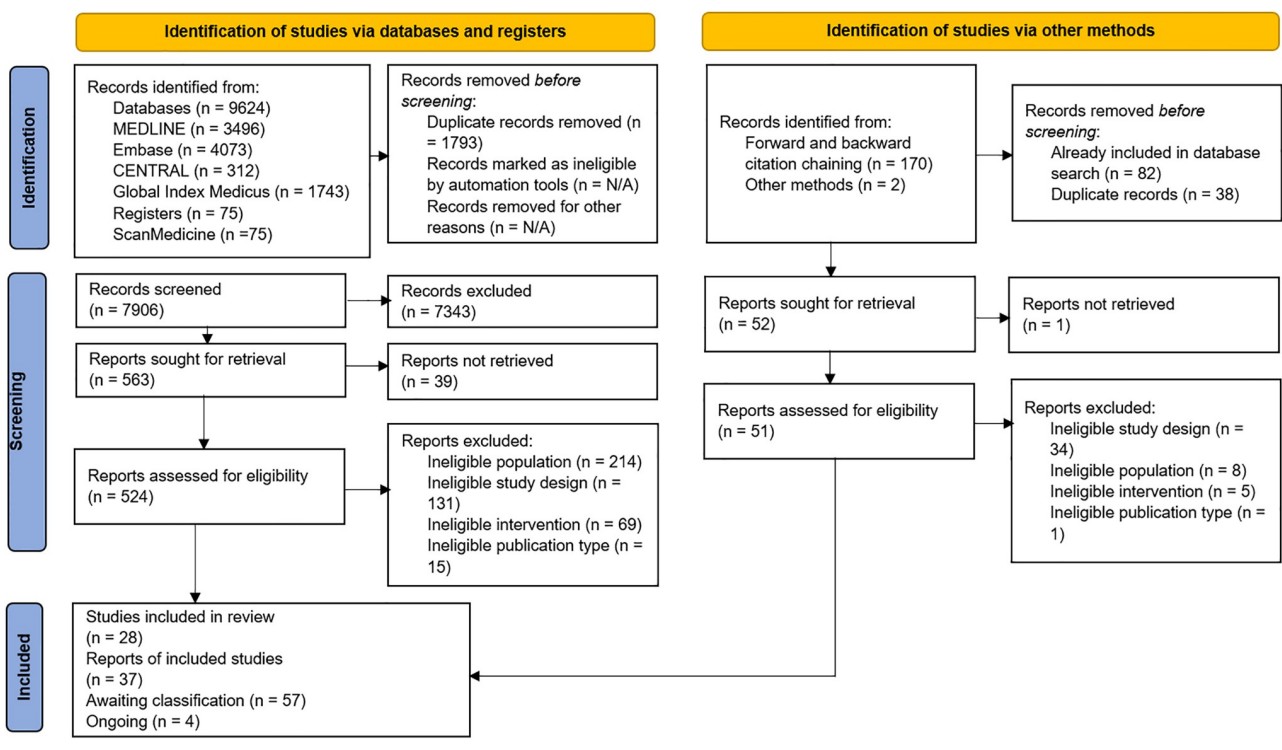

**Fig 3. PRISMA diagram.**

the JBI Checklist for Case Control Studies, and one systematic review was assessed using AMSTAR-2 [26–28]. Heat maps of risk of bias assessments across study types are contained within S9 Table. In brief, there were concerns about blinding of participants and personnel in RCTs [35, 51, 52], as well as a lack of accounting for confounders within cohort studies and the case-control study [31–33, 36, 40–44, 47, 49, 50]. The one included systematic review was deemed to be of moderate quality when assessed using AMSTAR-2 [26, 57].

**Evidence map results.** The completed evidence map can be seen in Fig 4.

**Interventions.** Interventions assessed by included studies are summarised in Table 5. Most studies (N = 20, 71%) assessed surgical interventions alone [31, 32, 34, 36, 38–44, 46–49, 53–56]. One study assessed the combined effects of surgery and catheterisation [58]. Given that 59% of survey respondents stated surgical interventions for treating obstetric fistula were most important to investigate, this suggests studies identified for the evidence map assess interventions respondents most wish to know about.

However, only one study examined psychological interventions [51], four examined catheterisation [33, 35, 52, 57], and one exclusively assessed the effects of physical therapy [45]. One study assessed the effects of bladder catheterisation, muscle training and surgery but as the method of surgery was not adequately described, only bladder catheterisation and muscle training could be included [50]. No studies assessed lifestyle interventions. Although these interventions were considered less important to the total group of respondents, more patients and members of the public considered psychological therapy, lifestyle interventions and physical therapy to be of greater importance to investigate further. It is therefore possible that the current evidence base may not address interventions of most interest to patients and members of the public; it is important to recognise however that only four patients and members of the public provided survey responses.

**Table 4. Characteristics of included studies (N = 28).**

|  | n | % |
|---|---|---|
| **Study design** |  |  |
| RCT | 7 | 25 |
| Prospective cohort | 6 | 21 |
| Retrospective cohort | 12 | 43 |
| Historical cohort | 1 | 4 |
| Case-control | 1 | 4 |
| Systematic review | 1 | 4 |
| **Setting** |  |  |
| Single centre | 16 | 57 |
| Multicentre | 10 | 36 |
| Not described | 1 | 4 |
| Not applicable (systematic review) | 1 | 4 |
| **Number of fistulae types included** |  |  |
| Single | 15 | 54 |
| Combination | 9 | 32 |
| Not described | 4 | 14 |
| **Type of fistulae included** |  |  |
| Vesicovaginal only | 13 | 46 |
| Rectovaginal only | 2 | 7 |
| Vesicovaginal only; rectovaginal only; both | 3 | 11 |
| Vesicovaginal only; both vesicovaginal and rectovaginal | 4 | 14 |
| Other | 2 | 7 |
| Not described | 4 | 14 |
| **Fistula classification system adopted** |  |  |
| Goh classification system | 6 | 21 |
| Waaldijk classification system | 3 | 11 |
| Other | 2 | 7 |
| Not described | 17 | 61 |
| **Study funding source described** |  |  |
| Yes | 10 | 36 |
| No | 18 | 64 |
| **Study conflicts of interest described** |  |  |
| Yes | 16 | 57 |
| No | 12 | 43 |

**Outcome measures.** Table 6 compares reported outcome measures within included studies, separated by intervention category. In terms of outcome measures survey respondents believed were most important, cure or improvement of obstetric fistula (24 studies, 85.7%) and cure or improvement in urinary incontinence (20 studies, 71.4%) were reported by many included studies, whilst other outcome measures were generally little reported. Only two studies (7.14%) reported on quality of life, the outcome considered most important by respondents [45, 53]. Similarly, improvement in faecal incontinence was only reported by two studies [47, 50]. Mental [51] and sexual health [32] were reported in a single study each. This suggests there is a large evidence gap surrounding the outcomes respondents most wish to know about.

**Sensitivity analysis.** The results of the sensitivity analysis are shown in Fig 5. Here, only seven of 28 studies were retained within the evidence map [37, 41, 43, 44, 46, 47, 52]. There are no longer any studies that assess the effects of psychological therapy or physical therapy, while

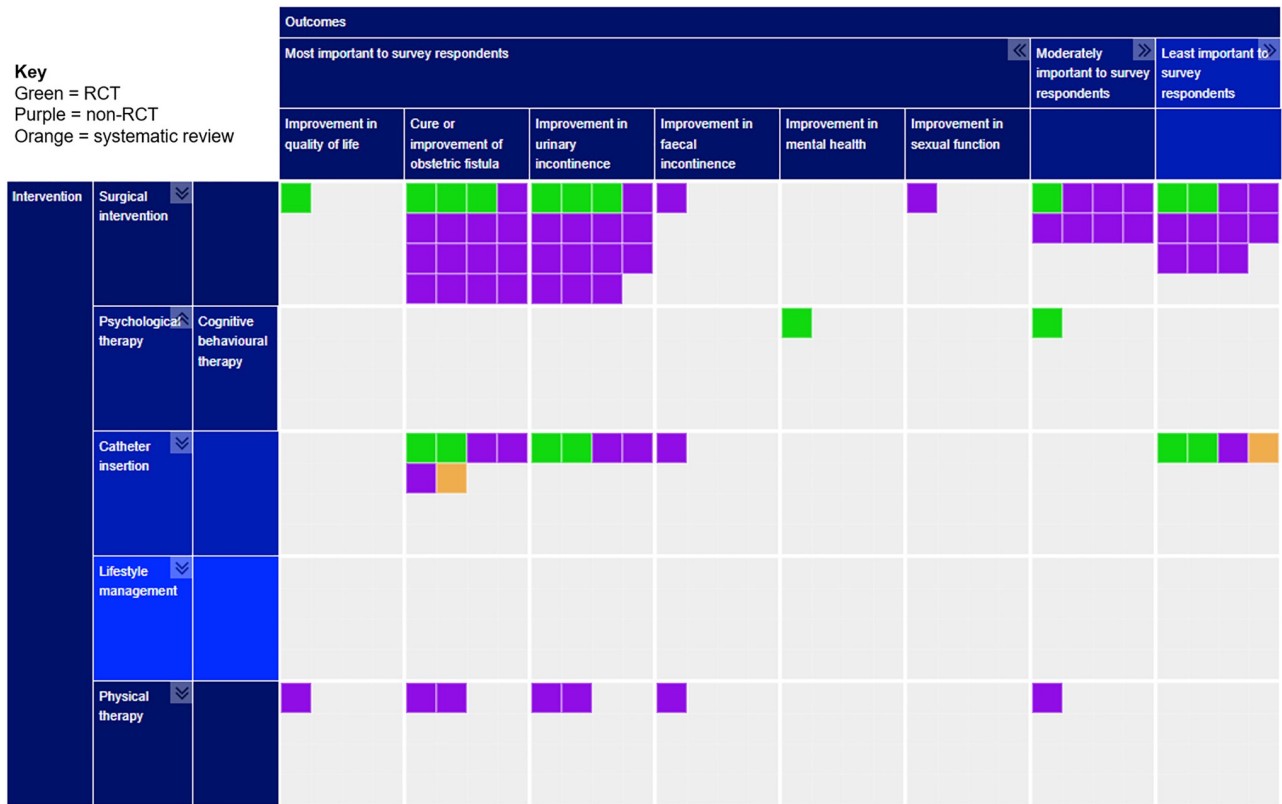

**Fig 4. Evidence map of interventions for treating obstetric fistula.** Key: Type of study: Green = RCT; Purple = Non-RCT; Orange = Systematic Review.

there are no longer any RCTs remaining that assess surgical interventions. As such, the sensitivity analysis demonstrates that the applicability of the evidence to women with obstetric fistula and the usefulness of the data in many studies is limited.

## Discussion

### Summary of main results

In total, 40 participants responded to the survey, of which 39 were analysed. Thirty-three reports of 28 studies were included in the evidence map. Most studies (71%) focused on surgical interventions for treating obstetric fistula, which aligned with what survey respondents most wanted to know, although no patient respondents wanted to know this. However, there was limited and occasionally no evidence surrounding other intervention categories and these interventions appeared to be most important to patients.

Only 2 of 28 studies reported on quality of life, the outcome of most importance to survey respondents [45, 53]. Improvement in mental health was only reported in a single study [51], as was sexual function [32]. This suggests there may be a gap in the evidence surrounding these outcomes. Improvement in faecal incontinence was only reported by two studies [47, 50]. It should be noted that faecal incontinence would only be relevant to studies of rectovaginal fistula, though as there were nine studies in total that included rectovaginal fistula, either alone or in combination with vesicovaginal fistula, arguably this still represents a gap in the current literature [32, 40, 41, 50, 51, 54, 58, 59]. Cure or improvement of obstetric fistula was

**Table 5. Interventions assessed by included studies (N = 28).**

| Intervention | N | % |
|---|---|---|
| **Surgical intervention alone** | | |
| Native tissue repair | 14 | 50 |
| Graft repair | 0 | 0 |
| Tissue flaps | 0 | 0 |
| Native tissue, tissue flaps or graft repair plus anal sphincter repair | 0 | 0 |
| Urinary diversion | 1 | 4 |
| Colostomy | 0 | 0 |
| Native tissue repair; tissue flaps | 4 | 14 |
| Native tissue, tissue flaps, sphincter repair, colostomy | 1 | 4 |
| **Psychological therapy alone** | | |
| CBT | 1 | 4 |
| **Catheter insertion alone** | | |
| Bladder catheterisation | 4 | 14 |
| Ureteral catheterisation | 0 | 0 |
| **Lifestyle intervention alone** | | |
| Skin protection | 0 | 0 |
| Pads | 0 | 0 |
| Urethral plugs | 0 | 0 |
| Vaginal lubricants | 0 | 0 |
| UTI prophylaxis | 0 | 0 |
| Dietary modification | 0 | 0 |
| **Physical therapy alone** | | |
| Therapeutic exercise | 0 | 0 |
| Bladder training | 0 | 0 |
| Bowel habit training | 0 | 0 |
| Muscle training | 1 | 4 |
| Co-ordination training | 0 | 0 |
| Biofeedback | 0 | 0 |
| Electrical muscle stimulation | 0 | 0 |
| **Combinations of intervention types** | | |
| Graft repair, tissue flaps and ureteral catheterisation | 1 | 4 |
| Bladder catheterisation, muscle training | 1 | 4 |

reported by 24 studies and cure or improvement in urinary incontinence was reported by 20 studies.

## Overall applicability of the evidence

Only 28 studies were eligible for the evidence map. Furthermore, sensitivity analysis demonstrated that only seven of 28 studies reported assessed interventions separately and had a population where at least 80% of the participants were confirmed to have obstetric fistula. This suggests the overall applicability of the evidence may be limited, as we cannot be certain whether the evidence is truly generalisable to women with the condition. It is possible that the Studies Awaiting Assessment, of which 52 were in a language other than English, could contribute more evidence but, due to pragmatic time constraints, it was not feasible to translate these records within this work.

**Table 6. Number of included studies reporting outcome measures.**

| Outcome | Survey responses (N = 155) | All studies (N = 28) | Surgical intervention (N = 20) | Psychologic therapy (N = 1) | Catheter insertion (N = 4) | Physical therapy (N = 1) | Multiple therapies (N = 2) |
|---|---|---|---|---|---|---|---|
| Improvement in quality of life | 25 | 2 | 1 | 0 | 0 | 1 | 0 |
| Cure or improvement in obstetric fistula | 24 | 24 | 17 | 0 | 4 | 1 | 2 |
| Cure or improvement in urinary incontinence | 24 | 20 | 15 | 0 | 3 | 1 | 1 |
| Cure or improvement in faecal incontinence | 17 | 2 | 1 | 0 | 0 | 0 | 1 |
| Improvement in mental health | 14 | 1 | 0 | 1 | 0 | 0 | 0 |
| Improvement in sexual function | 11 | 1 | 1 | 0 | 0 | 0 | 0 |
| Improvement in activities of daily living | 10 | 0 | 0 | 0 | 0 | 0 | 0 |
| Improvement in pain | 7 | 0 | 0 | 0 | 0 | 0 | 0 |
| Patient satisfaction with intervention | 5 | 3 | 2 | 1 | 0 | 0 | 0 |
| Woman's need for further treatment | 5 | 8 | 7 | 0 | 0 | 1 | 0 |
| Adherence to the intervention | 3 | 1 | 0 | 1 | 0 | 0 | 0 |
| Improvement in associated prolapse | 3 | 0 | 0 | 0 | 0 | 0 | 0 |
| Improvement in difficulties urinating | 2 | 2 | 0 | 0 | 1 | 0 | 1 |
| Improvement in urinary retention | 2 | 0 | 0 | 0 | 0 | 0 | 0 |
| Adverse events of the intervention | 2 | 12 | 10 | 0 | 2 | 0 | 0 |
| Improvement in difficulty defecating | 1 | 0 | 0 | 0 | 0 | 0 | 0 |
| Improvement in haematuria | 0 | 1 | 1 | 0 | 0 | 0 | 0 |

Other constraints on the applicability of the evidence are also apparent. Twenty-two studies were conducted within African countries [31–40, 45, 46, 48–50, 52–56, 58], two in the Middle East [41, 42], two in Asia [43, 44], and one in North America [47]. Although a meta-analysis has suggested that the prevalence of obstetric fistula is higher in Africa than South Asia (RR 1.60 versus RR 1.20 per 1000 people, respectively) [4], the evidence base as presented in this work may not reflect the burden on women in South Asian regions. How generalisable the findings are to this geographical context, and that of the Middle East, is unclear.

Furthermore, 13 studies exclusively examined the effects of interventions on vesicovaginal fistula [34–36, 38, 39, 42–44, 46, 53, 55–57], while four examined vesicovaginal fistula either alone or in combination with rectovaginal fistula [32, 40, 48, 54]. Only two studies examined rectovaginal fistula alone [41, 47], with three examining either rectovaginal, vesicovaginal, or a combination of both [50, 51, 58]. No included studies assessed women with urethrovaginal, vesico-uterine or ureterovaginal fistulae. Consequently, the generalisability of findings within the evidence map to women with rectovaginal, ureterovaginal, urethrovaginal and vesico-uterine fistulae is limited.

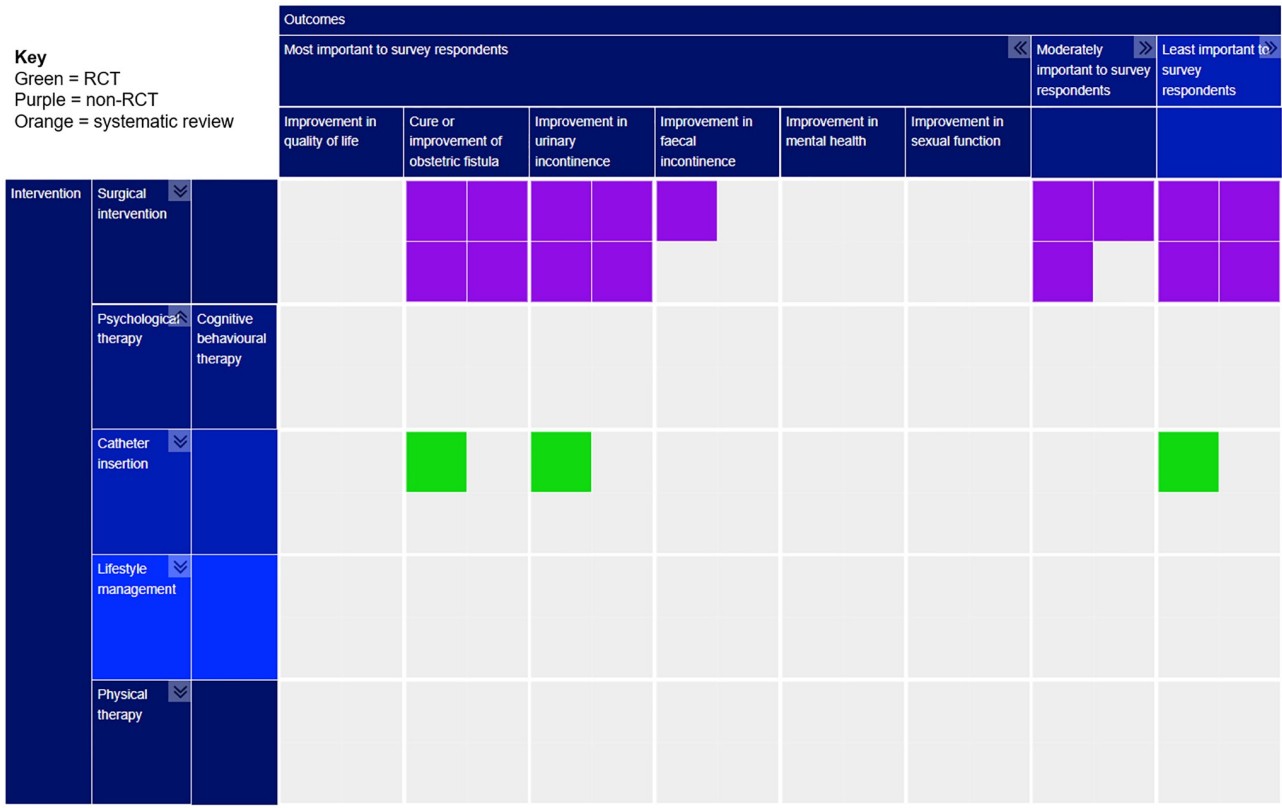

**Fig 5. Sensitivity analysis by proportion of participants with obstetric fistula and by whether studies reported interventions separately.** Key: Type of study: Green = RCT; Purple = Non-RCT; Orange = Systematic Review.

## Strengths and limitations

A strength of the survey that informed the review was the piloting undertaken to ensure functionality and comprehensibility. However, a limitation of the survey is the small number of responses, and the high proportion of respondents who were from high-income countries, despite our attempts to distribute it widely. This may limit the survey's applicability to different key stakeholders. Additionally, only four respondents were patients and none identified as immediate family members or partners of patients. Despite this, the use of the survey to inform the framework of the review and contextualise the findings is still a strength as it enhances its overall relevancy.

In so far as possible, the evidence map was conducted in accordance with Campbell Collaboration conduct and reporting standards [20, 29]. To ensure as many relevant studies were captured as possible, the search strategies were peer reviewed twice by an experienced Information Specialist using the PRESS guideline [23]. Additionally, no restrictions on language, type or timing of publication were enabled, multiple databases were used, and backwards and forwards citation chaining was conducted to ensure sensitivity. However, grey literature searching was not undertaken for this review, potentially resulting in omissions of small or unpublished studies [60]. However, given that this evidence map was a rapid synthesis of evidence, the Cochrane Rapid Reviews Method Group have noted that limiting to database searches and citation chaining is appropriate in these circumstances [61]. Additionally, study selection and data coding were mainly undertaken by a single reviewer, with a second checking a proportion; this contravenes gold standard guidance for traditional systematic reviews of

interventions [60, 62]. However, guidance from the Campbell Collaboration surrounding evidence map methodology states that double screening is not mandatory [17], while the Cochrane Rapid Reviews Methods Group notes that extraction can be undertaken by a single reviewer with a second checking 20% of records [61]. This approach was taken for this review, with additional methods planned to ensure quality control and consistency at both screening stages (see 'Study selection and data coding').

### Implications for patients and practitioners

Due to the paucity of research identified for this evidence map, there is currently little to guide patients and practitioners regarding what may be the most effective treatment options for women with obstetric fistula. Further research is required to guide informed choices by patients and practices undertaken by clinicians.

### Implications for research

It is imperative that robust, high-quality primary studies are conducted in the area to meet the WHO's goal of eradicating obstetric fistula by 2030 [63, 64]. This work identified four potentially relevant ongoing studies, all of which were RCTs [65–68]. However, research on treatments needs to ensure it is applicable to the population, by ensuring that most enrolled participants ($\geq$ 80% at a minimum) have fistula of obstetric aetiology. Similarly, the effects of different interventions, particularly surgical interventions, need to be reported separately and transparently within the results and employ robust, comparative study designs to facilitate future systematic reviews and meta-analyses. These studies should be reported in accordance with either the Consolidated Standards of Reporting Trials (CONSORT) or Strengthening the Reporting of Observational Studies in Epidemiology (STROBE) statements [69, 70].

As previously suggested [12], a core outcome set would also assist in standardising reporting across trials; these should be published and accessible through a platform such as the Core Outcome Measures in Effectiveness Trials (COMET) website [71]. The survey presented in this work gives an initial indication of what may be of most importance to respondents, though further work is required to ensure the perspectives of patients, the wider public and policy-makers are included in future research. Finally, in this work, only 12 of 27 included primary studies clearly stated they had adequate ethical clearance. Future primary studies should ensure ethical approval is granted and transparently reported so women with obstetric fistula are effectively safeguarded.

### Conclusions

This work has highlighted a paucity of robust evidence to inform patients, practitioners and policy-makers about interventions for treating obstetric fistula. Although there is some evidence surrounding different surgical interventions, there is currently little surrounding psychological interventions, catheter insertion, lifestyle interventions and physical therapies. Whilst there are some data on cure or improvement of obstetric fistula and cure or improvement in urinary incontinence there is little incorporating quality of life, faecal incontinence and improvement in sexual function. Significantly, much of the current research in the area is not truly applicable to women with obstetric fistula and reporting of intervention effectiveness is often not transparent. A core outcome set is needed to standardise measurements across future studies, informed by what patients and practitioners believe is most important to research. Further robust research is also needed to help address the evidence gaps identified by this work.

## Supporting information

**S1 Text. Differences between protocol and review.**
(DOCX)

**S2 Text. Finalised survey design.**
(DOCX)

**S1 Table. Campbell Collaboration reporting standards for evidence gap maps.**
(DOCX)

**S2 Table. Finalised search strategies.**
(DOCX)

**S3 Table. Full eligibility criteria for the evidence map.**
(DOCX)

**S4 Table. Coding tool for evidence map.**
(DOCX)

**S5 Table. List of excluded studies.**
(DOCX)

**S6 Table. Details of studies with multiple reports.**
(DOCX)

**S7 Table. Reasons for Studies Awaiting Classification.**
(DOCX)

**S8 Table. Characteristics of ongoing studies.**
(DOCX)

**S9 Table. Heat maps of risk of bias assessments across included studies.**
(DOCX)

## Acknowledgments

We would like to thank everyone who contributed to the distribution and completion of the survey. Additionally, we would also like to thank Sheila Wallace, Catherine Richmond and Alex Inskip for their support with the PRESS assessments and advice regarding search strategies.

## Author Contributions

**Conceptualization:** Eugenie Evelynne Johnson, Paul Hilton, Fiona Pearson, Judith Goh, Luke Vale.

**Data curation:** Eugenie Evelynne Johnson, Nicole O'Connor.

**Formal analysis:** Eugenie Evelynne Johnson.

**Funding acquisition:** Luke Vale.

**Methodology:** Eugenie Evelynne Johnson, Nicole O'Connor, Paul Hilton, Fiona Pearson, Luke Vale.

**Project administration:** Eugenie Evelynne Johnson.

**Supervision:** Fiona Pearson, Luke Vale.

**Validation:** Nicole O'Connor, Paul Hilton, Fiona Pearson, Judith Goh, Luke Vale.

**Visualization:** Eugenie Evelynne Johnson.

**Writing – original draft:** Eugenie Evelynne Johnson.

**Writing – review & editing:** Nicole O'Connor, Paul Hilton, Fiona Pearson, Judith Goh, Luke Vale.

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
