## [Decision Letter · Decision Letter 0]

21 Nov 2022

PGPH-D-22-01398

Interventions for treating obstetric fistula: an evidence gap map

Dear Dr. Johnson,

Thank you for submitting your manuscript to PLOS Global Public Health. After careful consideration, we feel that it has merit but does not fully meet PLOS Global Public Health’s publication criteria as it currently stands. Therefore, we invite you to submit a revised version of the manuscript that addresses the points raised during the review process.

We look forward to receiving your revised manuscript.

Kind regards,

Priyanka Baloni

Academic Editor

Journal Requirements:

2. Please send a completed 'Competing Interests' statement, including any COIs declared by your co-authors. If you have no competing interests to declare, please state "The authors have declared that no competing interests exist". Otherwise please declare all competing interests beginning with the statement "I have read the journal's policy and the authors of this manuscript have the following competing interests:"

3. Please amend your detailed Financial Disclosure statement. This is published with the article. It must therefore be completed in full sentences and contain the exact wording you wish to be published.

a. State what role the funders took in the study. If the funders had no role in your study, please state: “The funders had no role in study design, data collection and analysis, decision to publish, or preparation of the manuscript.”

b. If any authors received a salary from any of your funders, please state which authors and which funders.

4. Please provide separate figure files in .tif or .eps format only and remove any figures embedded in your manuscript file. Please also ensure that all files are under our size limit of 10MB.

5. We noticed that you used "unpublished data" in the manuscript. We do not allow these references, as the PLOS data access policy requires that all data be either published with the manuscript or made available in a publicly accessible database. Please amend the supplementary material to include the referenced data or remove the references.

6. We have noticed that you have uploaded Supporting Information files, but you have not included a list of legends. Please add a full list of legends for your Supporting Information files after the references list. 

Additional Editor Comments (if provided):

The reviewers have commented on the work. Overall, the topic is of great relevance and hope that the authors are able to address the reviewers comments and submit the revised version.

Reviewers' comments:

Reviewer's Responses to Questions

**Comments to the Author**

1. Does this manuscript meet PLOS Global Public Health’s publication criteria? Is the manuscript technically sound, and do the data support the conclusions? The manuscript must describe methodologically and ethically rigorous research with conclusions that are appropriately drawn based on the data presented.

Reviewer #1: Yes

Reviewer #2: Yes

2. Has the statistical analysis been performed appropriately and rigorously?

Reviewer #1: Yes

Reviewer #2: Yes

3. Have the authors made all data underlying the findings in their manuscript fully available (please refer to the Data Availability Statement at the start of the manuscript PDF file)?

Reviewer #1: Yes

Reviewer #2: Yes

4. Is the manuscript presented in an intelligible fashion and written in standard English?

Reviewer #1: Yes

Reviewer #2: Yes

5. Review Comments to the Author

Reviewer #1: Excellent choice of topic and a well-executed review. Despite the scope of the review being quite broad, the authors rationalized all every step, which contained the information from being too generalized.

The authors have mentioned the PICO framework. The population of interest was from LMICs. However, survey respondents and studies from HICs are also included. Please clarify.

'there is currently little to guide patients and practitioners regarding what may be the most effective treatment options for women with obstetric fistula' Surgical correction is scientifically proven to be the most effective treatment option. Even according to your own results (Survey 24/25 and surgical intervention 17/20) However, what is yet to established is a more holistic management approach. Such as adjunct therapies of surgery combined with psychological rehabilitation. So, I would suggest rephrasing this part for clarity.

'Although there is some evidence surrounding different surgical interventions, there is currently little surrounding psychological interventions, catheter insertion, lifestyle interventions and physical therapies.' I would suggest removing catheter insertion as it is a proven technique for treating early diagnosed, simple fistulas.

In implications or further research, I would suggest mentioning management and prevention of obstructed labor. As that is the most prevalent cause in LMICs and is avertible with better labor management.

Reviewer #2: In the manuscript "Interventions for treating obstetric fistula:an evidence gap map", Johnson et. al. address an important public health problem of obstetric fistula that impacts more than 50,000 women annually. The authors first survey stakeholders to guide the framework of creating an evidence map,followed by analysis of several quantitative primary studies and review to produce the first evidence map assessing interventions (surgical and non-invasive interventions) for treating obstetric fistula. The study highlighted a paucity of robust evidence to inform patients, practitioners and policy-makers about interventions for treating obstetric fistula, particularly with regard to non-surgical interventions.

Overall, this study is extremely relevant and valuable to the field of obstetric fistula research and it paves the way for future research in this area. The study is well designed and I believe the manuscript is suitable for publication if the authors address a few minor concerns:

1. My primary concern is the small sample size of patients (n=4) in the initial survey. While clinicians and academicians may provide a medical/statistical picture of the condition, the patients are the primary stakeholders here and it would be extremely important to have more patient responders. As the survey provides the foundation to structure outcomes for the evidence map, it becomes important to include more patients or perhaps weigh their responses higher than the rest. The authors have mentioned this factor in their limitations, but it is necessary to highlight this in the results section as well.

2. Similarly, the survey does not include spouses/partners and immediate family members of the patients. They would undoubtedly provide more insight as the condition has a significant impact on the patient’s social and community life, and family support does play an important role in guiding successful public health interventions.

3. The manuscript suffers from long length, and some sections should be condensed. Particularly, it would be important to emphasize the main findings regarding the lack of non-surgical interventions in current literature and practices - It would be useful to emphasize this in the abstract as well as in individual sections.

4. The authors mention the need for further robust research is also needed to help address the evidence gaps identified by their study. It would be useful to include the quantitative measures that future studies can include - for example, what would be the best way to evaluate improvements in mental health, and how perhaps improvement in quality of life can be assessed.

5. The figure legends are either missing (mostly titles are present) or are too short. Please add them and expand on each. Additionally, figure resolutions are not clear after conversion to pdf, the authors should take note of this.

6.Figures 4, 5a and 5b are not very intuitive to understand. While the key for the colors are mentioned in the legend, I would suggest the authors include the color key on top of the figures itself so that readers can easily interpret them.

7. Minor comment: On Line 122: “such as charities” is repeated twice

6. PLOS authors have the option to publish the peer review history of their article (what does this mean?). If published, this will include your full peer review and any attached files.

**Do you want your identity to be public for this peer review?** For information about this choice, including consent withdrawal, please see our Privacy Policy.

Reviewer #1: **Yes: **Dr. Samaa Akhtar

Reviewer #2: No

---

## [Decision Letter · Decision Letter 1]

20 Dec 2022

Interventions for treating obstetric fistula: an evidence gap map

PGPH-D-22-01398R1

Dear Miss Johnson,

We are pleased to inform you that your manuscript 'Interventions for treating obstetric fistula: an evidence gap map' has been provisionally accepted for publication in PLOS Global Public Health.

Best regards,

Priyanka Baloni

Academic Editor

Reviewer Comments (if any, and for reference):

Reviewer's Responses to Questions

**Comments to the Author**

1. If the authors have adequately addressed your comments raised in a previous round of review and you feel that this manuscript is now acceptable for publication, you may indicate that here to bypass the “Comments to the Author” section, enter your conflict of interest statement in the “Confidential to Editor” section, and submit your "Accept" recommendation.

Reviewer #1: All comments have been addressed

Reviewer #2: All comments have been addressed

2. Does this manuscript meet PLOS Global Public Health’s publication criteria? Is the manuscript technically sound, and do the data support the conclusions? The manuscript must describe methodologically and ethically rigorous research with conclusions that are appropriately drawn based on the data presented.

Reviewer #1: Yes

Reviewer #2: Yes

3. Has the statistical analysis been performed appropriately and rigorously?

Reviewer #1: Yes

Reviewer #2: Yes

4. Have the authors made all data underlying the findings in their manuscript fully available (please refer to the Data Availability Statement at the start of the manuscript PDF file)?

Reviewer #1: Yes

Reviewer #2: Yes

5. Is the manuscript presented in an intelligible fashion and written in standard English?

Reviewer #1: Yes

Reviewer #2: Yes

6. Review Comments to the Author

Reviewer #1: All comments have been addressed. Well done.

Reviewer #2: (No Response)

7. PLOS authors have the option to publish the peer review history of their article (what does this mean?). If published, this will include your full peer review and any attached files.

**Do you want your identity to be public for this peer review?** For information about this choice, including consent withdrawal, please see our Privacy Policy.

Reviewer #1: **Yes: **Dr. Samaa Akhtar

Reviewer #2: **Yes: **Awanti Sambarey
